# Beyond PD(L)-1 Blockade in Microsatellite-Instable Cancers: Current Landscape of Immune Co-Inhibitory Receptor Targeting

**DOI:** 10.3390/cancers16020281

**Published:** 2024-01-09

**Authors:** Edoardo Crimini, Luca Boscolo Bielo, Pier Paolo Maria Berton Giachetti, Gloria Pellizzari, Gabriele Antonarelli, Beatrice Taurelli Salimbeni, Matteo Repetto, Carmen Belli, Giuseppe Curigliano

**Affiliations:** 1Division of Early Drug Development, European Institute of Oncology, IRCCS, Via Giuseppe Ripamonti 435, 20141 Milan, Italygloria.pellizzari@ieo.it (G.P.); gabriele.antonarelli@ieo.it (G.A.);; 2Department of Oncology and Hemato-Oncology, University of Milan, 20122 Milan, Italy; 3Memorial Sloan Kettering Cancer Center, New York, NY 10065, USA

**Keywords:** cancer, microsatellite instability, immunotherapy, PD-L1, precision oncology, drug development, target therapy, combination treatment, biomarkers, agnostic

## Abstract

**Simple Summary:**

This review aims to describe the current state of the art in the field of immunotherapy applied to patients with cancer-harboring microsatellite instability, giving a comprehensive landscape of the possible therapeutical strategies in these patients. We discuss the potential of combination strategies and novel therapeutic approaches that may change the management of microsatellite-instable cancer in the future.

**Abstract:**

High microsatellite instability (MSI-H) derives from genomic hypermutability due to deficient mismatch repair function. Colorectal (CRC) and endometrial cancers (EC) are the tumor types that more often present MSI-H. Anti-PD(L)-1 antibodies have been demonstrated to be agnostically effective in patients with MSI-H cancer, but 50–60% of them do not respond to single-agent treatment, highlighting the necessity of expanding their treatment opportunities. Ipilimumab (anti-CTLA4) is the only immune checkpoint inhibitor (ICI) non-targeting PD(L)-1 that has been approved so far by the FDA for MSI-H cancer, namely, CRC in combination with nivolumab. Anti-TIM3 antibody LY3321367 showed interesting clinical activity in combination with anti-PDL-1 antibody in patients with MSI-H cancer not previously treated with anti-PD(L)-1. In contrast, no clinical evidence is available for anti-LAG3, anti-TIGIT, anti-BTLA, anti-ICOS and anti-IDO1 antibodies in MSI-H cancers, but clinical trials are ongoing. Other immunotherapeutic strategies under study for MSI-H cancers include vaccines, systemic immunomodulators, STING agonists, PKM2 activators, T-cell immunotherapy, LAIR-1 immunosuppression reversal, IL5 superagonists, oncolytic viruses and IL12 partial agonists. In conclusion, several combination therapies of ICIs and novel strategies are emerging and may revolutionize the treatment paradigm of MSI-H patients in the future. A huge effort will be necessary to find reliable immune biomarkers to personalize therapeutical decisions.

## 1. Introduction

Microsatellites are highly polymorphic repeated sequences of 1–6 nucleotides and are widespread in the human genome, accounting for 3% of it [1,2]. Due to their repetitive nature, these sequences are prone to mismatch errors caused by DNA polymerase slippage. The DNA mismatch repair system (MMR) is an evolutionarily conserved pathway able to repair these errors, ensuring the maintenance of genomic integrity [3]. Both acquired and inherited alterations and epigenetic modifications determining deficient-MMR (dMMR) function lead to the biological status known as microsatellite instability (MSI), characterized by changes in microsatellite base pairs [3]. MSI is defined as “high” (MSI-H) or “low” (MSI-L), respectively, when a threshold is or is not reached, depending on the panel used [4,5]. MSI-H/dMMR tumors show high sensitivity to ICIs because of their hypermutable phenotype and the higher neoantigen load [6].

To date, the methods of MSI detection are polymerase chain reaction (PCR) and next-generation sequencing (NGS), while immunohistochemistry (IHC) can assess the dMMR status [7]. IHC is an easy to perform and affordable and provides a widespread assay to detect dMMR, powered by the rapid evidence of results [8,9]. IHC’s main weaknesses are the reliance on the quality of tissue sample, the analysis of a limited number of proteins, and the unreliability in non-truncating mutations leading to a loss of function of MMR proteins without a loss of expression [8,9]. Instead, PCR-based methods detect all types of mutation and are less influenced by pre-analytical issues. The main disadvantage is that normal tissue needs to be co-tested, while availability and costs are sustainable. Finally, NGS assays allow the simultaneous assessment of many biomarkers but are limited by the high amount of tissue required, the costs and the longer turnaround time [10]. To clarify, the American Society of Clinical Oncology (ASCO) endorsement of the American Pathologist Guidelines was published, with different testing options allowed, according to the cancer types [11].

MSI-H/dMMR is far more common in CRC and EC cancers [12]. At first, the efficacy of pembrolizumab was demonstrated in pre-treated MSI-H/dMMR CRC patients [13]. Subsequent evidence of the magnitude of benefit provided by pembrolizumab when compared to standard chemotherapy in the first-line setting for patients affected by MSI-H/dMMR CRC led it to be considered a new standard of care [14]. In addition, nivolumab with or without ipilimumab was approved as the second line in the MSI-H/dMMR CRC population [15]. Instead, the efficacy of dostarlimab was consistent only in the MSI-H/dMMR EC cohort of the GARNET trial [16]. First-line applications are still under evaluation.

MSI-H/dMMR was the first tumor-agnostic biomarker approved by the US Food and Drug Administration (FDA) [17]. Since 2017, any previously treated advanced MSI-H/dMMR solid tumor (2–4% of all cancers) may receive pembrolizumab (an anti-PD1 IgG4 monoclonal antibody) in the United States of America (USA) based on the results of the KEYNOTE-158 trial [18]. KEYNOTE-158 was designed only for non-colorectal cancers (non-CRC) and met its primary endpoint with an objective response rate (ORR) of 34.3% [18]. Out of 233 patients, 27 different tumor types were represented; the most frequent were EC, gastric, cholangiocarcinoma, pancreatic and small intestine cancers [18]. Dostarlimab, another anti-PD1 IgG4 antibody, was the second ICI approved as a tumor-agnostic drug by FDA [17]. Its efficacy was evaluated in 209 patients affected by recurrent or advanced solid tumors with MSI-H/dMMR who progressed to systemic therapy and had no alternative treatment [19]. Cohort F explored the solid MSI-H/dMMR tumors (except EC and non-NSCLC), among which the most common histology was CRC, followed by small intestine, gastric, pancreatic and ovarian cancers. Globally, the primary endpoint ORR was met (41.6%) [19].

On the whole, while the role of anti-PD(L)-1 ICIs is well established in the treatment of MSI-H/dMMR solid tumors, few data are actually available regarding the potential benefit of drugs targeting other co-inhibitory receptors in this subset of oncological patients. Figure 1 contains a timeline showing the milestones in the employment of ICIs in the treatment of MSSI-tumors.

In this review, we provide an overview of the current evidence on this topic.

## 2. Immune Checkpoint Molecules

CTLA-4 (cytotoxic T-lymphocyte-associated protein 4) is a co-inhibitory receptor primarily found on the surface of activated T cells [20]. CTLA-4 regulates immune responses by binding to B7-1 (CD80) and B7-2 (CD86), co-stimulatory ligands found on antigen-presenting cells, disrupting their binding with CD28. CD28 represents a key co-stimulatory receptor of T-cells, and its affinity for B7-1 and B7-2 is lower compared to CTLA-4. Thus, when co-expressed, CTLA-4 abates T cell activation [20,21]. Aside from binding to B7 proteins, which represents the most important mechanism exerted by CTLA4, it reduces T cell activation also thanks to the transendocytosis of B7 from the antigen-presenting cells (APCs), the suppression of T-cell receptor signaling and the disruption of the c-SMAC (central supramolecular cluster) within the immunological synapse [22]. CTLA-4 upregulation has been described as a mechanism for immune-evasion of cancer cells [23].

The TIM (T-cell immunoglobulin domain and mucin domain) superfamily includes surface glycoproteins that regulate immunologic processes [24]. Four TIM members have been identified in humans (Tim-1, Tim-2, Tim-3 and Tim-4) [24]. Tim-1, like Tim-4, is primarily involved in repair processes, particularly regarding apoptotic bodies clearance [25]. Tim-2 is mainly expressed in TH2 cells and inhibits TH1 responses [26]. Tim-3 exerts regulatory functions for both innate and acquired immune responses, is mainly expressed on NK cells and dysfunctional T cells and is frequently co-expressed in the latter with PD-1 [27,28]. Moreover, it is expressed by dendritic cells, in which it was shown to impair the response to nucleic acid ligands for toll-like receptors 3, 7 and 9 and suppressing HMGB1-mediated recruitment of nucleic acids to endosomes [29]. In macrophages, Tim-3 acts as a negative regulator of the NLRP3 inflammasome [29]. Tim-3 has been observed to be upregulated in tumor cells progressing to treatment with anti-PD-1/PD-L1 in preclinical and clinical models, and thus, is regarded to play a key role in cancer immunoescape [30,31].

TIGIT (T cell immunoreceptor with Ig and ITIM domains) is a cell surface receptor protein expressed on various types of immune cells, including T cells, NK cells and Tregs [32,33]. TIGIT binds to several ligands, including CD155 (poliovirus receptor) and CD112 (nectin-2), which are both expressed on antigen-presenting cells and tumor cells [32,33]. TIGIT contains immunoreceptor tyrosine-based inhibition motifs (ITIMs) in its cytoplasmic domain, which recruit inhibitory enzymes, such as phosphatases, to prevent T cell activation [33]. TIGIT regulates immune responses, particularly by suppressing antitumor immunity and promoting Treg function [32,33].

BTLAs (B- and T-lymphocyte attenuators) are inhibitory receptors expressed on several immune cells, including T and B cells, NK cells and dendritic cells [34]. BTLA’s ligand herpesvirus entry mediator (HVEM) activates tyrosine phosphorylation of the tyrosine-based inhibitory motif (ITIM) and recruits the protein tyrosine phosphatases, SHP-1 and SHP-2, which mediate immunosuppressive effects [35]. On the other side, binding of the Grb-2 association motif with Grb-2 recruits PI3K protein subunit p85, leading to T cell activation [35]. In tumor cells, the binding of BTLA to HVEM inhibits T cell activation and proliferation [34,35]. BTLA has been shown to be upregulated in T cell-infiltrating tumors and taking part in suppressing anti-tumor immune responses [35].

ILT3 (Immunoglobulin-like transcript 3) is a cell surface receptor expressed by a variety of immune cells, including dendritic cells, macrophages, and some T and B cells [36]. ILT3 was demonstrated to have the physiological function of rendering APCs tolerogenic [37]. ILT3 binds to a ligand called HLA-G, which has been shown to be expressed by cancer cells and is thought to contribute to its ability to evade the immune system. By binding to HLA-G, ILT3 suppresses the immune response against cancer cells and promotes tumor growth [38,39,40].

LAG-3 (Lymphocyte-activation gene 3) is an inhibitory receptor expressed on activated T cells, regulatory T cells, and natural killer (NK) cells [41]. Its main function is to regulate the immune response by binding to major histocompatibility complex (MHC) class II molecules on antigen-presenting cells, such as dendritic cells, macrophages, and B cells [41,42]. LAG-3 is thought to play a role in limiting T-cell responses and preventing excessive inflammation [41,42]. In cancer, LAG-3 is often upregulated in tumor-infiltrating lymphocytes and is associated with T cell exhaustion [43].

VISTA (V-domain Ig suppressor of T cell activation) is a cell surface receptor protein expressed on various types of immune cells, including T cells, B cells, dendritic cells, and myeloid cells [44]. VISTA has been shown to play a role in regulating immune responses by inhibiting T cell activation and promoting the development of Tregs, which are important for controlling excessive immune responses [44]. In cancer, VISTA has been observed to be upregulated on tumor cells and immune cells within the tumor microenvironment, where it contributes to immune evasion and tumor progression [45].

B7-H3 (B7 homolog 3 protein), also known as CD276, is a member of the B7 family of immune checkpoint proteins [46]. It is a type I transmembrane protein expressed on various immune cells, including dendritic cells, macrophages and T cells, as well as on non-immune cells, such as cancer cells [46]. B7-H3 is involved in regulating immune responses by interacting with an unknown receptor on T-cells [46,47]. Its role in cancer is not fully understood, but it has been found to be overexpressed in various types of tumors and is thought to play a role in promoting tumor growth and suppressing antitumor immune responses [48,49].

## 3. Clinical Evidence on Co-Inhibitory Receptor Targeting in MSI-H Cancers

In this section, we will provide a thorough discussion of the completed clinical trials regarding patients with MSI-H tumors treated with anti-CTLA4, anti-TIM, anti-LAG and anti-TIGIT, alone or in combination with other drugs, highlighting the most relevant results.

### 3.1. CTLA4

Ipilimumab was the first anti-CTLA4 ICI approved by FDA for the treatment of MSI-H cancers in 2018. Ipilimumab demonstrated its efficacy in the CheckMate142 phase II trial [50] (NCT02060188), in which 119 patients with advanced dMMR/MSI-H CRC that progressed after one or more lines of systemic chemotherapy received nivolumab (3 mg/kg) and low-dose ipilimumab (1 mg/kg), followed by nivolumab maintenance. BRAF mutation has been found in 25% of patients. At a median follow-up of 50.9 months (range 46.9–62.7 months), an objective response was achieved in 65% of the patients (95% CI 55–73%), with a disease control ≥12 weeks in 81% of the patients (95% CI 72–87%). No new safety signals were reported. Moreover, CheckMate142 also included a cohort for patients with MSI-H CRC in the first line setting, whose results were published in October 2021 [51]. Nivolumab and ipilimumab showed an ORR of 69% (95% CI, 53 to 82%), while the median duration of response (mDOR), median progression-free survival (mPFS) and median overall survival (mOS) were not reached. These encouraging results led to the ongoing phase III CheckMate 8HW trial (NCT04008030), which is testing Nivolumab, Nivolumab Plus Ipilimumab or Investigator’s Choice Chemotherapy in patients affected by MSI-H CRC.

On the other side, anti-PD(L)1 and anti-CTLA4 combination therapies are also being tested in the early setting. One of most intriguing studies is the phase-2 NICHE clinical trial (NCT03026140), whose final analysis was published in 2022 [52]. In this study, patients with non-metastatic, resectable dMMR or pMMR colon cancer received a single dose of anti-CTLA4 ipilimumab (at a dosage of 1 mg/kg) combined with two doses of anti-PD1 nivolumab (3 mg/kg) and underwent surgery within 6 weeks. Patients with pMMR were also randomized to receive celecoxib. The key primary endpoint was safety, and the major secondary endpoints included a pathologic response (defined as 50% or less viable tumor rest, VTR and a major pathologic response (MPR) defined as <10% VTR) and disease-free survival (DFS). Of thirty-two patients evaluable for efficacy in the dMMR cohort, a pathologic response was observed in 32/32 patients, with 31/32 MPR (97%, 95% CI 91–100%) and one partial response. A pathologic complete response (pCR) was observed in 22/32 (69%, 95% CI 53–85%) patients. None of the dMMR patients had disease recurrence. These data have been confirmed in the phase-2 NICHE-2 trial [53] (NCT04165772), where 112 patients with dMMR locally advanced colon cancer who received ipilimumab and nivolumab with the same schedule as in the NICHE trial had a pathologic response in 99% of cases (106/107 of evaluable patients), with 95% MPR and no disease recurrence at a median follow-up of 13 months (range 1–57). In the INFINITY multi-cohort phase-2 trial [54] (NCT04817826) presented at ASCO GI 2023, 18 patients with resectable gastric or gastroesophageal junction adenocarcinoma MSI-H enrolled in cohort 1 received a 12-week treatment of anti-CTLA4 tremelimumab (300 mg single dose) and durvalumab 1500 mg q4 weeks for 3 cycles, followed by surgery, with a pCR rate as the primary endpoint. Among the evaluable patients, the pCR rate was 60% (9/15) and MPR was 80% (12/15). At a median follow-up of 13.4 months (range 9.7–14.2 months), 2 progressive diseases and 4 deaths occurred (1 related to progressive disease and 2 to surgical complications). The authors performed an exploratory analysis and noted that the PD-L1 combined positive score (CPS) was not associated with the outcomes.

### 3.2. TIM-3

In a phase 1b trial (NCT02791334), 82 patients with MSI-H advanced solid tumors were enrolled to receive the anti-PDL1 LY3300054 monotherapy (40 patients, all naïve to previous anti-PDL1) or the combination treatment with LY3300054 and the anti-TIM3 LY3321367 (42 patients, 20 PD-L1 naïve and 22 progressing to prior anti-PDL1) [55]. The LY3300054 monotherapy resulted in 32.5% ORR (13/40) and a 60.0% (24/40) disease control rate (DCR) [55]. Five patients (12.5%) had a complete response. The mPFS for the monotherapy cohort was 7.4 months (95% CI 1.8–23.8 months) [55]. In the combination therapy cohort, the ORR and DCR in the PD-L1 naïve patients were 45.0% (9/20) and 70.0% (14/20), respectively, with two patients exhibiting a complete response (10%) lasting for 12.8 and 14.8 months [55]. The mPFS was 7.6 months (95% CI 1.9 months-NR). In the combination therapy with previous anti-PDL1 exposure, one patient (4.5%) had a partial response and six patients (27.3%) had a stable disease. The PD-L1 tumor proportion score was not associated with the response in any cohort [55].

### 3.3. LAG

In a phase II trial (NCT03607890), 15 patients with dMMR tumors that progressed during an anti-PD1 therapy were enrolled in cohort 1 to receive anti-PD1 Nivolumab 480 mg + anti-LAG3 Relatlimab 160 mg q4w [56]. The interim analysis results published in 2022 showed that of 13 patients evaluable with a median follow-up (mFUP) of 12.4 months, there were 5 patients with a stable disease (SD), 1 with a partial response (PR) and 1 with a complete response (CR) [56]. No safety concerns were raised (trAEs occurred in 6 patients and were all grade 1–2) [56].

Figure 2 contains a representation of the mechanisms of action of the immune checkpoint molecules for which clinical data are available.

### 3.4. TIGIT

Very limited data are available on the clinical impact of the anti-TIGIT drugs in MSI-H tumors. The phase I trial NCT02794571, employing tiragolumab and atezolizumab, enrolled a patient with MSI-H CRC in phase Ia. The patient received tiragolumab 1200 mg with prolonged SD and then crossed to the phase Ib with tiragolumab 600 mg and atezolizumab 1200 mg with a prolonged PR (>45 months) [57]. No clinical data are actually available for vibostolimab in patients with MSI-H tumors, but the ongoing trial KEYSTEP-008 has a cohort with vibostolimab + pembrolizumab enrolling patients with MSI-H CRC [58].

## 4. Ongoing Trials Enrolling Patients with MSI-H Cancers

Different trials are currently underway to investigate co-inhibitory strategies in dMMR/MSI-H tumors (Table 1).

Following the results of the single-arm phase II CheckMate-142 trial, CheckMate 8HW (NCT04008030) is currently testing nivolumab alone or in combination with ipilimumab, as compared with physician-choice chemotherapy, as a first-line regimen for dMRR/MSI-H CRC, which could establish a new standard immunotherapy beyond pembrolizumab as a first-line regimen for d-MMR/MSI-H CRC, further exploring the possible synergistic effect of a co-inhibitory strategy.

The MOST-CIRCUIT is a phase II, multi-cohort trial testing ipilimumab plus nivolumab among dMMR/MSI-H advanced solid tumors (excluding CRC) that progressed to standard therapies, with ORR and 6-month PFS as the co-primary endpoints.

In addition to mAbs, bispecific antibodies represent a novel treatment strategy targeting two epitopes of the same or different antigens. In this regard, cadonilimab (AK104) is a bispecific antibody targeting PD-1 and CTLA-4 that is currently being tested in dMRR/MSI-H CCR both in early disease (NCT04556253; NCT05815290) and in the metastatic setting among pretreated CCR (NCT05426005).

Beyond CTLA-4, several trials are currently underway to target additional co-inhibitory receptors. Concerning dMMR/MSI-H tumors, the TIGIT-directed mAb tiragolumab is being tested in combination with atezolizumab in pretreated dMRR/MSI-H advanced solid tumors in the multi-arm, phase II TIRACAN trial (NCT05483400).

In the previously reported multi-arm phase II NICHE-3 trial (NCT03026140), relatlimab, a mAb targeting LAG-3, is being examined in combination with nivolumab in two distinct cohorts of pMMR/MSS and dMMR/MSI CRR to investigate whether the blockade of other co-inhibitory receptors beyond CTLA-4, in addition to PD-1, could provide similar outcomes. Moreover, the same combination of relatlimab plus nivolumab is being tested in metastatic dMMR/MSI-H solid tumors, which progressed to prior immune checkpoint therapy with PD-1/PD-L1 (NCT03026140).

Another interesting trial is the KEYSTEP-008, which is investigating pembrolizumab in combination with different checkpoint inhibitors (CTLA-4, LAG-3, TIGIT and ILT-4) in patients with MSI-H CRC [58].

## 5. Other Immunotherapeutic Strategies for MSI-H Cancers

### 5.1. Tumor Microenvironment

Another way to improve the anti-cancer response is through acting on the tumor microenvironment (TME) eliciting or inhibiting several molecules that are involved in the immune response [59].

The cyclic GMP-AMP synthase (cGAS)/stimulator of the interferon genes (STING) pathway has a primary role in the activation of an innate immune response to pathogenic micro-organisms. Usually, this pathway is activated by the presence of microbial DNA, but even self-DNA leaked from the nucleus or mitochondria and related genomic instability represent an initial trigger [60]. Indeed, the cGAS–STING pathway seems to have a dual role in cancer immunity, both promoting cell growth through modulation of the tumor microenvironment and otherwise activating anti-tumoral inflammatory response, particularly via induction of tumor-specific CD8^+^ T cells [61,62,63]. There are preclinical data that suggest elevated STING expression could induce PD-L1 expression and is related to a good response to ICIs; therefore, it is a potential way to enhance the response to ICIs in MSI-H tumors [64,65].

Several agonists of the STING pathway have been studied, such as cyclic dinucleotide (CDN) analogues, but their clinical application is limited by their huge molecular weight, low stability and permeability [66]. Indeed, they mostly need intratumoral administration. However, agonists of STING have been evaluated in phase 1 and 2 trials, but patients are not selected by MSI status [67].

The pyruvate kinase (PK) is an enzyme that converts phosphoenolpyruvate to pyruvate in the glycolysis process, but while the M1 isoenzyme (PKM1) is found in adult cells, the M2 variant (PKM2) is active in embryonic and cancer cells [68]. PKM2 fundamentally promotes glycolysis in tumor tissue, but even it has a transcriptional role in eliciting proliferation and cancer cell survival [68,69,70]. Interestingly, PKM2 seems to also have a function in tumor immune escape, since it favors the transformation from TAM1 to TAM2 and the overexpression of PD-L1 in cancer cells [71].

Therefore, dimer inhibition or tetramer activation of PKM2 could potentially reduce cancer growth.

It should be noted that only pre-clinical data are available regarding inhibitors and activators of PKM2, as reviewed by Chhipa et al. [70]. The challenge of finding an effective anti-cancer molecule depends on the fact that PKM2 seems to have different roles in distinct malignancies and PKM1 has a pro-oncogenic role when PKM2 is inhibited [70]. A phase 1 trial (NCT04328740) is recruiting patients with advanced solid tumor, metastatic renal cell carcinoma (RCC), MSI-H mCRC and non-small cell lung cancer (NSCLC) to receive TP-1454 as monotherapy or in combination with nivolumab and ipilimumab. TP-1454 is a PKM2 activator and increases its tetramer formation, but it mostly favors the immune response decreasing T-reg cells in in vitro models [72].

Leukocyte-associated immunoglobulin-like receptors (LAIR)-1 and LAIR-2, members of the leukocyte receptor complex (LRC), have a role in causing immune escape. Indeed, LAIR-1 is a co-inhibitory receptor expressed by several immune cells that limits T cell activation and, additionally, it favors adhesion to collagens [73]. The increased expression of LAIR-1 in cancer cells elicits immune escape, reducing T cell activity and supporting an immune-excluded TME with the production of collagens [73]. LAIR-1 was blocked by LAIR-2, a secreted protein; thus, a molecule that acts like LAIR-2 could potentially remodel the TME and change a cold tumor into a hot one. Pre-clinical data showed that NC410, a dimeric form of the LAIR-2 protein fused to a human Fc domain of the immunoglobulin (Ig) subtype IgG1, may elicit T cell activation and potentiate the anti-tumor effect of anti ICIs therapy [74,75]. Indeed, a phase I/II study has recruited patients with metastatic solid tumors and tested NC410, but the results have not yet been published (NCT04408599). However, the limitation of studying similar molecules is the difference between the TME in human tissue compared to that in in vitro and animal models.

### 5.2. Cytokines and Cytokine Superagonists

The administration of several cytokines has been studied for treating tumors by a non-specific immune stimulation; however, the response rates for cytokine monotherapy are moderate and often limited by dose-related toxicity. Indeed, the true value of cytokines may lie in their synergistic effects when combined with other immunotherapies or using modified cytokines [76,77].

IL15 is a cytokine that stimulates T cells and natural killer (NK) cells, eliciting antibody-dependent cellular cytotoxicity [78,79]. However, IL15 has a short half-life, and an IL15 superagonist was designed to use this cytokine for clinical aims with more potency and stability [80]. Indeed, N-803 (also known as Anktiva), consists of three parts: the cytokine (IL-15N72D), the cytokine fusion (IL-15Rα) and the antibody (IgG1 Fc). It has been demonstrated to selectively stimulate NK cells/CD8-positive T cells in pre-clinical models and clinical activity in several solid tumors (NCT03228667) [81,82,83], especially in BCG-unresponsive CIS and papillary non-muscle-invasive bladder cancer (NMIBC) (NCT03022825) (NCT02138734) [84,85]. In addition, N-803 is being investigated in phase 2/3 trials for other solid tumors in combination with chemotherapy, (NCT04390399) anti-PD1 (NCT03520686), ramucirumab (NCT05096663) and Sacituzumab govitecan (NCT04927884). Thus, it is a very promising drug, but data for a specific subtype of cancers, such as MSI tumors, are missing. Another promising molecule is SOT101, which contains IL15 and the Sushi+ domain of IL-15Rα; since it does not require dendritic cells to present IL15, it is extremely active in stimulating an immune response [86]. The safety and clinical activity of this drug were demonstrated in the phase 1/2 trial AURELIO-03 (NCT04234113), both as monotherapy and in combination with pembrolizumab, in several solid tumors comprehensive of MSI-H tumors [86,87,88]. Since the majority of patients had a clinical benefit, the AURELIO-04 trial (NCT05256381) is recruiting patients with selected solid tumors, such as unresectable or metastatic MSI-H/dMMR CRC, to further evaluate the efficacy of SOT101 in combination with pembrolizumab [89].

### 5.3. Bispecific T-Cell Engagers

Another approach is the use of bispecific T-cell engager (BiTE) antibodies. These antibodies are created by combining two different single-chain antibodies, resulting in a new antibody that has a variable region capable of recognizing tumor antigens and another region that can bind to the T-cell marker CD3, triggering T cell activation [90]. However, a potential drawback of this approach is that it lacks T-cell specificity since CD3 is expressed not only by cytotoxic T cells, but also by Treg cells. One example of a BiTE antibody currently being tested in a Phase 1b clinical trial (NCT03866239) is cibisatamab (CEATCB), which is specific for CEA. Investigators evaluated it in combination with atezolizumab but in patients with MSS CRC [91].

### 5.4. Other T Cell Immunotherapies

Investigators have recruited patients with MSI-H solid tumors as well as epithelial ovarian, fallopian tube or peritoneal cancer, hepatocellular carcinoma, NSCLC and urothelial cancer to receive DPX-Survivac in combination with low-dose cyclophosphamide and pembrolizumab in a phase 1 trial (NCT03836352). DPX-Survivac represents an innovative T cell immunotherapy utilizing the DPX platform to stimulate potent and persistent survivin-specific T cell reactions against cancerous cells. By necessitating active uptake of antigens by antigen-presenting cells at the injection site, DPX aims to enhance the duration and strength of targeted immune responses. Several trials recruiting patients with ovarian cancer demonstrated that this molecule elicits an important anti-tumor response (NCT01416038; NCT03332576; NCT02785250) [92].

### 5.5. Oncolytic Viruses

Intriguingly, oncolytic virus is a recent therapy under investigation in combination with ICIs. An oncolytic virus may have an anti-cancer role both directly, by infecting the cancer cells and leading them to cell lysis, and indirectly, by exposing neoantigens from the lysed cell and thus eliciting the immune response toward the tumor [93,94]. Their potential role in enhancing the activity of ICIs depends on the release of antigens that activate immune effectors, while ICIs enable these effectors to function efficiently [95,96,97,98]; hence an oncolytic virus may turn a cold TME into a hot TME, favoring the activity of ICIs in tumors that have demonstrated poor clinical benefits, such as MSI-H mCRC [97,99]. Indeed, several oncolytic viruses are under investigation for pMMR/MSS/MSI-L metastatic CRC [100].

The appealing point of oncolytic viruses is that they can be the foundation for engineering armed viruses that can enhance the effectiveness of ICIs, for example, by carrying the PD-1/PD-L1 antibody gene or other genes, which enable enhanced anti-PD-1 treatment, such as IL-7 or IL-12 codifying genes, as reviewed by Ren et al. [100]. Nevertheless, clinical data are still poor, and the limitations include the occurrence of immune-related adverse events (irAEs), mostly in combination therapy, and the multiplicity of antigenic expression by cancer cells that also has spatial and temporal variability. Another limit is the reproducibility of immune responses from animal models to patients, especially when the combination with ICIs is tested.

### 5.6. Vaccines

Another way to elicit an immune response is by modulating it using vaccines. Vaccines show promise in the context of MSI-H cancers since they exhibit high mutation rates, resulting in the generation of several neoantigens. In dMMR cells, uncorrected mismatches manifest as insertion/deletion mutations. Consequently, the immunogenicity of dMMR cancers arises not only from the sheer quantity of somatic mutations, but also from the vast number of potential epitopes within frameshift peptide sequences (FSP) triggered by insertion/deletion mutations [101,102]. Consequently, peptides generated from frameshifted mutations may be more effective as targets for vaccines compared to proteins derived from point mutations [103,104]. By specifically targeting these unique antigens, vaccines have the potential to stimulate a robust immune response against MSI-H cancer cells. Further research and clinical trials are underway to explore the effectiveness of vaccines in MSI-H cancer treatment and prevention [105].

A recent phase I/II trial investigated an FSP vaccine that encompasses peptides derived from the frameshift sequence of the coding microsatellite (cMS)-bearing genes TAF1B, HT001 and AIM2 in patients with dMRR CRC. All the patients had an immune response, but according to RECIST criteria, the tumor response could be assessed in only three patients. Two (66.6%) of these patients showed a stable disease as the best overall response [106].

Fakih et al. assessed a polyvalent viral vectored vaccine named Nous-209, encoding 209 shared FSPs among dMMR/MSI-H cancers, in combination with pembrolizumab in patients with mCRC, gastric and gastroesophageal cancers (NCT04041310). The phase 1 trial showed clinical activity and good tolerance of the vaccine investigated; however, the mPFS and mDoR were not reached (NCT01461148) [107].

Another phase 1 trial, active but not recruiting, is evaluating the toxicity of vaccination with frameshift-derived neoantigen-loaded DC of MSI CRC patients and persons who are known to be carriers of a germline MMR-gene mutation with no signs of disease (NCT01885702).

Table 2 contains a summarization of this section.

Therefore, evidence for immune activation with alternative immune-modulating strategies is available for MSI-H solid tumors, but the data on an actual improvement in clinical outcomes are still limited. There are still numerous challenges in the aspects of vaccine design, including a selection of antigens, delivery systems, adjuvants and vaccine types.

## 6. Discussion and Future Perspectives

The actual landscape of immunotherapeutic agents for MSI-H solid tumors is in rapid evolution. Different strategies are under study, but the large majority of them encompasses the combination of anti-PD(L)-1 with other immunomodulatory agents. In fact, the treatment of MSI-H solid tumors actually cannot disregard the administration of anti-PD(L)-1 drugs due to the striking and agnostic efficacy demonstrated by this class of drugs [13,14,16,18,19,108].

In this perspective, there are actually two major settings for the development of novel immunotherapeutic agents targeting co-inhibitory receptors other than PD(L)-1 or modulating immune response: their combination with anti-PD(L)-1 and the development of novel strategies after the failure of anti-PD(L)-1 treatment.

In the first condition, the strongest evidence arises from the combination of anti-CTLA4 + anti-PD(L)-1, the only combination that has thus far demonstrated an improvement of survival outcomes in patients with MSI-H cancers [51]. The other compounds under study still need clinical validation, but different classes of drugs have shown promising activity, including vaccines and oncolytic viruses. Nevertheless, there are still numerous challenges in the aspects of vaccine design, including a selection of antigens, delivery systems, adjuvants and vaccine types.

The difficulty in really understanding which is the added benefit of co-inhibitory receptor targeting relies on their association with different anti-PD(L)-1 drugs, some targeting PD1 and others PD(L)-1. A meta-analysis published in 2022 by Xiang et al. on the performance of different immunotherapeutic regimens in different cancer types showed that cemiplimab was the drug that impacted most positively the PFS of patients, while nivolumab and pembrolizumab impacted OS [109]. These results indicate that anti-PD(L)1 drugs may have different activity themselves, complicating the interpretation of combination trial results.

Nevertheless, relatlimab showed encouraging results even in the PD(L)-1-resistant MSI-H solid tumors, possibly widening the treatment possibilities of these patients, who often do not have valid therapeutic alternatives after anti-PD(L)1 failure due to the intrinsic chemoresistance of MSI-H neoplasms [56].

On the whole, there is a strong need for identifying new biomarkers predicting the benefit of immunotherapy and possibly of specific immunotherapeutic agents, even within the MSI-H tumors. The recurrent evidence arising from clinical trials, despite the employment of different ICIs directed toward different immune-checkpoint, is the unreliability of PD-L1 expression as an agnostic biomarker predictive of response in MSI-H solid tumors [110].

In the perspective of a rapid increase of treatment options for patients with MSI-H cancers and, in general, for all oncological patients, thanks to the streamlining of bureaucracy surrounding clinical trials and to the application of artificial intelligence tools for patient enrollment, clinical trial management and real-world data mining, an effort is required to assess which biomarkers should be analyzed in clinical trials testing specific classes of drugs, such as PD-L1, MSI/MMR, TILs, POLE mutations, TMB for immunotherapeutic agents, antigen expression levels for monoclonal antibodies, bispecific antibodies and antibody-drug conjugates, the mutational status of the target and of other genes associated with resistance for tyrosine-kinase inhibitors. This approach would lead to a better definition of the subset of patients to treat with each drug and ease cross-trial comparison of drug activity.

To conclude, novel inhibitors targeting immune checkpoints other than PD(L)1 are showing promising results in the treatment of MSI tumors. The results of ongoing clinical trials, especially the ones employing combination treatments, are eagerly awaited.

## Figures and Tables

**Figure 1 cancers-16-00281-f001:**
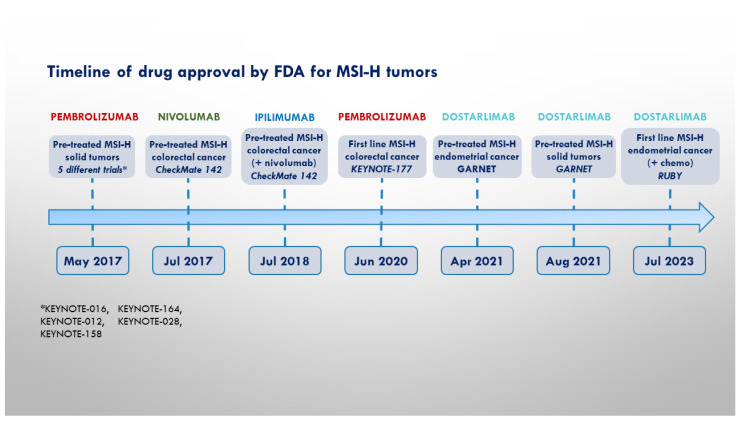
Timeline of pivotal trials and FDA approval for MSI-H cancers.

**Figure 2 cancers-16-00281-f002:**
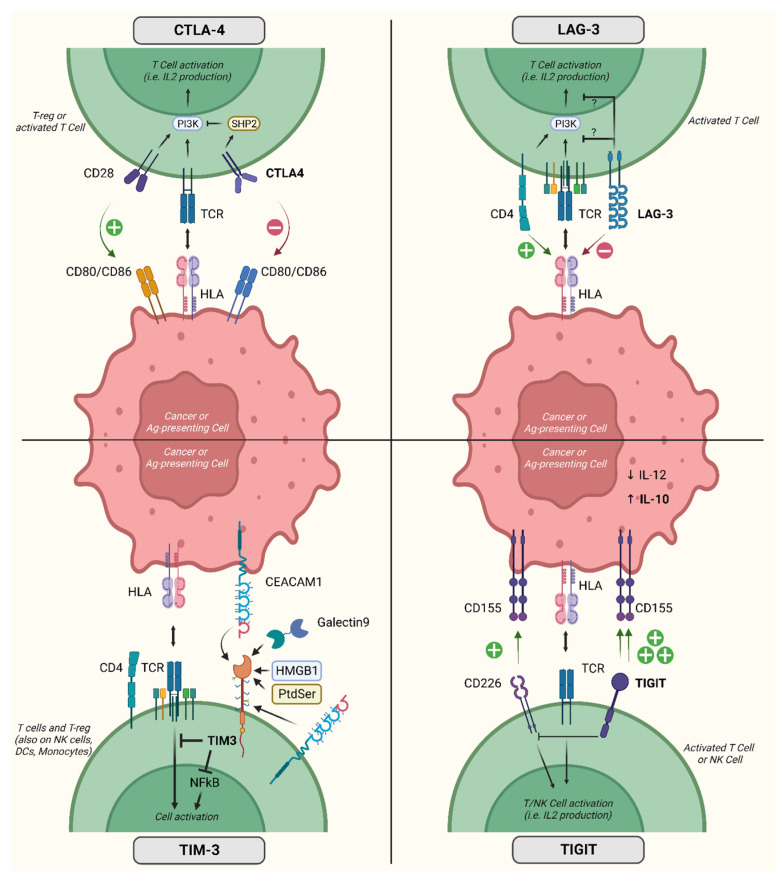
Activity and mechanism of action of relevant inhibitory co-receptors. Acronyms: CTLA-4, cytotoxic T-lymphocyte-associated protein 4; LAG-3, lymphocyte-associated gene 3; TIM-3, T cell immunoglobulin domain and mucin domain; TIGIT, T-cell immunoreceptor with immunoglobulin and ITIM domains; IL, interleukin; T-reg, regulatory T cells; CEACAM1, carcinoembryonic antigen-related cell adhesion molecule 1; CD, cluster of differentiation; HLA, human leukocyte antigen; TCR, T cell receptor; Ag, antigen; NFkB, nuclear factor kappa-light-chain-enhancer of activated B cells; HMGB1, high mobility group box 1 protein; NK, natural killer; DCs, dendritic cells; PI3K, phosphatidylinositol-3 kinase; PtdSer, phosphatidylserine; SHP2, Src homology phosphatase 2. Created with BioRender.com (accessed on 31 May 2023).

**Table 1 cancers-16-00281-t001:** Phase II and phase III clinical trials investigating immunotherapy co-inhibitory strategies in solid tumors with and without MSI-H/dMMR profile.

Agent	Target	Trial	Ph.	No.	Tumor Type	Setting	Study Treatment	Endpoint
Ipilimumab	**CTLA4**	**NCT04969887 (MOST-CIRCUIT) ***	2	240	Cohort 4: dMMR/MSI-H AST	M+ pretreated	Ipilimumab + Nivolumab	ORR; 6-month PFS
		**NCT04730544 ***	2	96	dMRR/MSI-H CCR	M+ 1-2L	Ipilimumab + Nivolumab	AEs; PFS
		**NCT04008030 (CheckMate 8HW) ***	3	831	dMRR/MSI-H CCR	Part 1: M+ pretreated Part 2: M+ 1L	A: Nivolumab B: Nivolumab + Ipilimumab C: ICC	PFS
Quavonlimab	**CTLA4**	**NCT04895722 (KEYSTEP-008) ***	2	320	MSI-H CRC	M+ pretreated (cohort A); M+ untreated (cohort B)	Cohort A Pembrolizumab vs. Quavonlimab/pembrolizuma; Cohort B: Pembrolizumab vs. Quavonlimab/Pembrolizumab vs. Favezelimab/Pembrolizumab vs. Vibostolimab/pembrolizumab vs. MK-4830 + pembrolizumab	ORR

Cadonlimab	**bsAb PD-1/CTLA-4**	**NCT05426005**	2	25	dMRR/MSI-H CCR	M+ pretreated	Cadonlimab	ORR
		**NCT04556253**	2	29	dMRR/MSI-H CCR and dMRR/MSI-H gastric cancer	Perioperative	Cadonlimab	pCR
		**NCT05815290**	2	50	dMRR/MSI-H CCR	Neoadj	Cadonlimab	pCR
Ociperlimab	TIGIT	NCT04746924	3	660	NSCLC	M+ pretreated. PD-L1 ≥ 50%	A: Ociperlimab + Tislezumab B: Pembrolizumab C: Tislezumab	PFS; OS
		NCT04866017	3	700	NSCLC	Stage III following CRT	A: Ociperlimab + Tislelizumab B: Tislelizumab C: Durvalumab	PFS
		NCT04732494	2	120	ESCC	M+ 2L PD-L1 ≥ 10%	A: Ociperlimab + Tislelizumab B: Tislelizumab	ORR
		NCT05023109	2	45	BTC	M+ 1L	Ociperlimab + Tislelizumab + Gemcitabine + Cisplatin	ORR
		NCT05014815	2	270	NSCLC	M+ 1L	Ociperlimab + Tislelizumab + chemotherapy	PFS
Tiragolumab	TIGIT	NCT04543617 (SKYSCRAPER-07)	3	750	ESCC	Following CRT	A: Tiragolumab + Atezolizumab B: Atezolizumab C: Placebo	PFS; OS
		NCT04294810 (SKYSCRAPER-01)	3	660	NSCLC	M+ 1L PD-L1 ≥ 50%	A: Tiragolumab + Atezolizumab B: Atezolizumab	PFS; OS; AEs
		NCT05805501	2	210	RCC	M+ 1L	Tiragolumab + RO7247669 (bsAb PD1-LAG3) + Axitinib	PFS
		NCT03708224	2	55	HNSCC	Neoadj	Tiragolumab + Atezolizumab + Tocilizumab	≥40% increase in infiltrating CD3; R0 resection rate
		NCT05009069	2	76	Rectal cancer	Following Neoadj CRT	A: Adj Tiragolumab + Atezolizumab B: Adj Atezolizumab	pCR
		**NCT05483400 * (TIRACAN)**	2	97	Cohort A: HNSCC Cohort B: AST MSI-H Cohort C: melanoma	Cohort A: neoadj Cohort B: M+ pretreated Cohort C: M+ pretreated	Tiragolumab + Atezolizumab	Cohort A: Pcr Cohort B and C: ORR
Domvanalimab (AB154)	TIGIT	NCT05568095 (STAR-221)	3	970	Gastric, GEJ or esophageal adenocarcinoma	M+ 1L	A: Domvanalimab + Zimberelimab + Oxaliplatin + 5-FU + Capecitabine B: Nivolumab + Oxaliplatin + 5-FU + Capecitabine	OS
		NCT04736173 (ARC-10)	3	750	NSCLC	M+ 1L PD-L1 ≥ 50%	A: Carboplatin + Pemetrexed + Paclitaxel B: Zimberelimab C: Domvanalimab + Zimberelimab D: Pembrolizumab	OS
		NCT04791839	2	30	NSCLC	Cohort A: M+ 1L PD-L1 1–49% Cohort B: M+ 1L PD-L1 ≥ 50%	Domvanalimab + Zimberelimab + Etrumadenant (A2R antagonist)	ORR
		NCT05130177	2	26	Melanoma	M+ ≥ 1L progressed on PD-1	Domvanalimab + Zimberelimab	ORR
Vibostolimab	TIGIT	NCT05665595 (MK-7684A-010/KEYVIBE-010)		1560	Melanoma	Adj	A: Vibostolimab + Pembrolizumab B: Pembrolizumab	RFS
		**NCT04895722 (KEYSTEP-008) ***	2	320	MSI-H CRC	M+ untreated (cohort B)	Cohort B: Pembrolizumab vs. Quavonlimab/Pembrolizumab vs. Favezelimab/Pembrolizumab vs. Vibostolimab/pembrolizumab vs. MK-4830 + pembrolizumab	ORR
HLX301	bsAb TIGIT-PD-L1	NCT05102214	1/2	150	AST	M+ pretreated	HLX301	AEs; DLTs, RP2D; ORR; DCR
		NCT05390528	1/2	30	AST or Lymphoma	M+ pretreated	HLX301	AEs
M6223	TIGIT	NCT05327530	2	252	Bladder	M+ pretreated	Cohort C: Avelumab + M6223	PFS
Fianlimab	LAG3	NCT05352672	3	1590	Melanoma	M+ 1L	A: Fianlimab + Cemiplimab B: Pembrolizumab C: Cemiplimab	PFS
		NCT05608291	3	1530	Melanoma	Adj	A: Fianlimab + Cemiplimab B: Pembrolizumab	RFS
INCAGN02385	LAG3	NCT05287113	2	162	HNSCC	M+ PD-L1 ≥ 1%	A: Retifanlimab B: Retifanlimab + INCAGN02385 C: Retifanlimab + INCAGN02385 + INCAGN02390	PFS
		NCT04586244	2	45	Bladder	Neoadj cisplatin inelegible	Cohort D: INCAGN02385 + Retifanlimab	CD8 increase
Eftilagimod Alpha	MHC II agonist	NCT05747794 (AIPAC-003)	2/3	849	Breast	TNBC 1L; HR+/HER2- ≥ 2L	A: Eftilagimod Alpha + Paclitaxel B: Paclitaxel	DLTs; AEs; OS
		NCT04811027	2	154	HNSCC	Cohort A and B: M+ 1L PD-L1 CPS ≥ 1% Cohort C: M+ 1L PD-L1 CPS < 1%	A: Eftilagimod alpha + Pembrolizumab B: Pembrolizumab C: Eftilagimod alpha + Pembrolizumab	ORR
Relatlimab	LAG3	NCT04080804	2	60	HNSCC	Neoadj	A: Relatlimab + Nivolumab B: Nivolumab + Ipilimumab C: Nivolumab	AEs
		NCT05134948	1/2	24	AST	M+ pretreated	Relatlimab + Nivolumab	AEs
		NCT05704647	2	30	Melanoma	M+ pretreated with active brain metastases	Relatlimab + Nivolumab	AEs
		NCT04095208	2	67	Soft-tissue sarcoma	M+ pretreated	Relatlimab + Nivolumab	ORR
		NCT04095208	2	61	Skin squamous cell carcinoma	1L	Relatlimab + Nivolumab	ORR
		NCT03642067	2	96	CRR	≥2L	Relatlimab + Nivolumab	ORR
		NCT03607890 *	2	42	AST	M+ MSI-H resistant to PD1/PD-L1 therapy	Relatlimab + Nivolumab	ORR
		NCT04205552 (NEOpredict)	2	90	NSCLC	Neoadj	A: Nivolumab B: Nivoluamb + Relatlimab	AEs
		**NCT03026140 * (NICHE-3)**	2	268	CRC	Neoadj Cohort 5: pMRR/MSS; Cohort 6: dMRR/MSI-H	Cohort 5: Relatlimab + Nivolumab Cohort 6: Relatlimab + Nivolumab	AEs, DFS
Favezelimab	LAG3	**NCT04895722 (KEYSTEP-008) ***	2	320	MSI-H CRC	M+ untreated (cohort B)	Cohort B: Pembrolizumab vs. Quavonlimab/Pembrolizumab vs. Favezelimab/Pembrolizumab vs. Vibostolimab/pembrolizumab vs. MK-4830 + pembrolizumab	ORR
RO7247669	bsAb PD1-LAG3	NCT05805501	2	210	RCC	M+ 1L	RO7247669 (bsAb PD1-LAG3) + Tiragolumab + Axitinib	PFS
		NCT04140500	1/2	320	AST	M+ pretreated	RO7247669	AEs; DLT; ORR; DCR
		NCT05805501	2	210	RCC	M+ pretreated	Cohort A: RO7247669 + Axitinib Cohort B: RO7247669 + Tiragolumab + Axitinib	PFS
HLX26	LAG3	NCT05787613	2	60	NSCLC	M+ 1L	HLX26 + Serplulimab + Pemetrexed + Nab-paclitaxel + Carboplatin	DLT; MTD; ORR
TSR-022	TIM-3	NCT03680508	2	42	HCC	M+ 1L	A: TSR-022 + TSR-042	ORR
		NCT04139902	2	56	Melanoma	Neoadj	A: TSR-022 + Dostalimab B: Dostarlimab	MPR
AZD7789	bsAb PD1-TIM3	NCT04931654	1/2	81	NSCLC	M+ ≥ 2L; part B2 IO-naive	A: AZD7789	AEs, DLT, ORR
Lomvastomig	bsAb PD1-TIM3	NCT04785820	2	210	Esophageal SCC	M+ ≥ 2L	A: Lomvastomig B: Tobemstomig (bsAb PD1-LAG3) C: Nivolumab	OS
INCAGN02390	TIM-3	NCT05287113	2	162	HNSCC	M+ PD-L1 ≥ 1%	A: Retifanlimab B: Retifanlimab + INCAGN02385 C: Retifanlimab + INCAGN02385 + INCAGN02390	PFS
		NCT04586244	2	45	Bladder	Neoadj	Cohort E: INCAGN02385 + Retifanlimab + INCAGN02390	CD8 increase
TQB2618	TIM-3	NCT05645315	1/2	127	Cohort 1: solid tumors Cohort 2: NSCLC	Cohort 1: pretreated MC Cohort 2: 1L NSCLC PD-L1 ≥ 1%	A: TQB2618 + TQB2450	DLT; ORR
		NCT05563480	2	90	Nasopharyngeal carcinoma	Part 1: M+ ≥ 2L Part 2: M+ 1L	A: TQB2618 + Pempulimab + Chemotherapy B: Penpulimab + Chemotherapy C: TQB2618 + Pempulimab	MTD; ORR; PFS
JS004	BTLA	NCT05664971	1/2	240	NSCLC; SCLC	Cohort 1-3: NSCLC M+ 1-3L; Cohort 4: SCLC M+ 1L	Cohort 1: JS004 + Toripalimab Cohort 2: JS004 + Toripalimab + Docetaxel Cohort 3: JS004 + Toripalimab + Pemetrexed + Cisplatin or Carboplatin Cohort 4: JS004 + Toripalimab + Carboplatin + Etoposide	AEs; SAEs; ORR
		NCT04929080	1/2	149	Nasopharyngeal carcinoma, HNSCC	M+ ≥ 2L	JS004 alone or + Toripalimab	AEs; ORR
KVA12123	VISTA	NCT05708950	1/2	314	AST	M+ pretreated	KVA12123 alone or + Pembrolizumab	AEs; RP2D
MK-4830	ILT-4	**NCT04895722 (KEYSTEP-008) ***	2	320	MSI-H CRC	M+ untreated (cohort B)	Cohort B: Pembrolizumab vs. Quavonlimab/Pembrolizumab vs. Favezelimab/Pembrolizumab vs. Vibostolimab/pembrolizumab vs. MK-4830 + pembrolizumab	ORR

Abbreviations: Ph., phase; No., number; MSI-H/dMMR, microsatellite instability high/mismatch repair deficient; CRC, colorectal cancer; ICC, investigator’s choice chemotherapy; pCR, pathological complete response; NSCLC, non-small cell lung cancer; M+, metastatic; PFS, progression-free survival; OS, overall survival; ESCC, esophageal squamous cell carcinoma; CRT, chemoradiotherapy; RCC, renal cell carcinoma; BTC, biliary tract cancer; GEJ, gastroesophageal junction; 5-FU, 5-fluorouracil; Adj, adjuvant; RFS, relapse-free survival; HR+/HER2-, HR-positive/HER2-negative; HCC; hepatocellular carcinoma; Neoadj, neoadjuvant. MPR, major pathologic response; ORR, overall response rate; DCR, disease control rate; DLT, dose-limiting toxicity; AEs, adverse events; IO, immuno-oncology; SCC, squamous cell carcinoma; HNSCC, head and neck squamous cell carcinoma; pMMR/MSSproficient mismatch repair (pMMR)/microsatellite stable; MC, metastatic cancer; MTD, maximum tolerated dose; SCLC, small-cell lung cancer; SAEs, serious adverse events; AST, advanced solid tumors; RP2D, recommended Phase 2 dose. * and bold: trials specifically including MSI-H/dMMR tumors as distinct cohorts.

**Table 2 cancers-16-00281-t002:** Summary of immunotherapeutic strategies described in Section 5.

Class	Target/Mechanism of Action	Example Drugs	Tumor Types	Trials
Tumor microenvironment	STING agonists	E7766	Solid tumors and lymphomas	NCT04144140
		Cyclic dinucleotide analogues	Solid tumors	-
	PKM2 inhibitors	TP-1454	Solid tumors, MSI-H CRC	NCT04328740 *
	LAIR-1 inhibitors	NC410	Solid tumors	NCT04408599 NCT05572684
Cytokine superagonists	IL15 superagonist	N-803	Solid tumors, various combination therapies with chemotherapy, anti-VEGF, antibody-drug conjugates, anti-PD1	NCT03228667 NCT03022825 NCT02138734 NCT04390399 NCT03520686 NCT05096663 NCT04927884
		SOT-101 Monotherapy or + anti-PD1 SOT101 + anti-PD1	Solid tumors	AURELIO-03 (NCT04234113) AURELIO-04 (NCT05256381) *
Bispecific T-cell engagers	CEA + CD3	Cibisatamab	MSS-CRC	NCT03866239
Oncolytic viruses	Cancer cells with release of antigens	RP1 +/− anti-PD1	Solid tumors	NCT03767348 *
Vaccines	Various cancer antigens	MSI-induced FSPs combined with MONTANIDE ISA-51	dMMR cancers	NCT01461148 *
		Nous-209	dMMR/MSI-H cancers	NCT04041310 *
Other T cell immunotherapies	Induction of survivin-specific T cell reactions	DPX-Survivac	Solid tumors, MSI-H tumors	NCT03836352 *

* trials specifically including MSI-H/dMMR tumors as distinct cohorts.

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
