# Peer review of "Beyond PD(L)-1 Blockade in Microsatellite-Instable Cancers: Current Landscape of Immune Co-Inhibitory Receptor Targeting"

_cancers, 2024, doi:10.3390/cancers16020281_

Round 1

Reviewer 1 Report

Comments and Suggestions for Authors

The authors wants describe the current state of the art in the field of immuno
therapy applied to patients with cancer harboring microsatellite instability. The review is written quite well but some changes are needed:

Lines 18-19: add the acronym for these cancer types (CRC/EC line 63), as they will be used later.

Lines 21, 177, 178, 189, 191, 213, 227, 274: specify the acronym of ICI, mDOR, mPFS, …

Line 38: (1), (2) -> (1, 2)…

Lines 63-70: better introduce the ICIs an, as done in the abstract, add which ICIs the antibodies are directed towards

Line 93: Co-inhibitory receptor, perhaps it would be better to write immune checkpoint molecules

Lines 94-100-111-119-125-131-138: make the writing homogeneous, or all:  CTLA-4 (cytotoxic T-lymphocyte-associated protein 4) or T-cell immunoglobulin domain and mucin domain (TIM)…

Line 107: myeloid?

Lines 93-151: Is it possible to indicate the physiological function of the different receptors for each ICI?

Line 145: B7-H3 -> B7 homolog 3 protein

Line 208: TIM -> TIM-3

Lines 210-212: the anti-PDL1 LY3300054 monotherapy (n=40 anti–PD-1/PD-L1–naïve patients), or the combination treatment with anti-PDL1 LY3300054 and the anti-TIM3 LY332136720  (n=20 anti–PD-1/PD-L1–naïve patients and n=22 progressed after prior PD-1/PD-L1 inhibitor treatment patients) -> UNCLEAR

Line 307: in combination with what?

Figure 2: it is not mentioned in the text, why are only CTLA-4/ LAG-3/TIM-3 / TIGIT considered relevant?

Table 1: add a column for CTLA-4, TIGIT…, they are not clearly visible

Add a table to resume other immunotherapeutic strategies for MSI-H cancers

Author Response

Dear Reviewer, thank you for your valuable comments, that will improve the quality of the manuscript. Here the change made to match your review:

Lines 18-19: add the acronym for these cancer types (CRC/EC line 63), as they will be used later.

Acronym added here and removed from line 63.

Lines 21, 177, 178, 189, 191, 213, 227, 274: specify the acronym of ICI, mDOR, mPFS, …

Acronyms specified.

Line 38: (1), (2) -> (1, 2)…

Unfortunately, the bibliography formatting style of MDPI requires separate brackets.

Lines 63-70: better introduce the ICIs an, as done in the abstract, add which ICIs the antibodies are directed towards.

We specified targets and IgG class: From 2017, any previously treated advanced MSI-H/dMMR solid tumor (2-4% of all can-cers) may receive pembrolizumab (an anti-PD1 IgG4 monoclonal antibody) in the United States of America (USA), based on the results of KEYNOTE-158 trial [18]. KEYNOTE-158 was designed only for non-colorectal cancers (non-CRC), and met its primary endpoint with an objective response rate (ORR) of 34.3% [18]. Out of 233 patients, 27 different tumor types were represented, the most frequent were EC, gastric, cholangiocarcinoma, pancre-atic, and small intestine cancers [18]. Dostarlimab, another anti-PD1 IgG4 antibody, was the second ICIs approved as a tumor-agnostic drug by FDA [17].

Line 93: Co-inhibitory receptor, perhaps it would be better to write immune checkpoint molecules

Changed accordingly.

Lines 94-100-111-119-125-131-138: make the writing homogeneous, or all:  CTLA-4 (cytotoxic T-lymphocyte-associated protein 4) or T-cell immunoglobulin domain and mucin domain (TIM)…

We homogenised the text indicating the acronym first for all the molecules.

Line 107: myeloid?

According to what is reported in the reference 26 (Han et al.): “Tim-3 was found to be involved in the expansion of myeloid-derived suppressor cells”. Moreover, it is expressed also by macrophages, even if it is most known to be expressed by lymphoid cells. We reformulated that section to be clearer and to better specify the physiological function of the receptor, as per following comment.

Lines 93-151: Is it possible to indicate the physiological function of the different receptors for each ICI?

Thank you for the comment. We shortly indicated the physiological function of the different receptors. We would not deepen excessively to remain focused on the clinical perspective of the paper.

Line 145: B7-H3 -> B7 homolog 3 protein

Changed accordingly.

Line 208: TIM -> TIM-3

Changed accordingly.

Lines 210-212: the anti-PDL1 LY3300054 monotherapy (n=40 anti–PD-1/PD-L1–naïve patients), or the combination treatment with anti-PDL1 LY3300054 and the anti-TIM3 LY332136720  (n=20 anti–PD-1/PD-L1–naïve patients and n=22 progressed after prior PD-1/PD-L1 inhibitor treatment patients) -> UNCLEAR

We modified the sentence to improve readability: In a phase 1b trial (NCT02791334) 82 patients with MSI-H advanced solid tumors were enrolled to receive the anti-PDL1 LY3300054 monotherapy (40 patients, all naïve to previous anti-PDL1) or the combination treatment with LY3300054 and the anti-TIM3 LY3321367 (42 patients, 20 PD-L1 naïve and 22 progressing to prior anti-PDL1).

Line 307: in combination with what?

Specified the combination with nivolumab and ipilimumab.

Figure 2: it is not mentioned in the text, why are only CTLA-4/ LAG-3/TIM-3 / TIGIT considered relevant?

We decided to focus the figure only on the molecules for which clinical data are available to improve readability. We moved the figure after the corresponding section and mentioned it in the text, and added a paragraph on TIGIT among the receptors with clinical evidence.

Table 1: add a column for CTLA-4, TIGIT…, they are not clearly visible.

We added a column with drug’s target.

Add a table to resume other immunotherapeutic strategies for MSI-H cancers

We added table 2 to summarize the section 5 (other immunotherapeutic strategies) and we divided the section into paragraphs to improve readability, as suggested by another reviewer.

Reviewer 2 Report

Comments and Suggestions for Authors

This review focuses on emerging combination therapies of ICIs and novel immunological strategies for microsatellite-instable cancers. The review is undoubtedly relevant and will interest a wide range of readers of the journal.

My recommendation and suggestions.

1) To improve readability, Section 5 “Other immunotherapeutic strategies for MSI-H cancers” would benefit from being divided into subsections.

2) Section 7 “Conclusions” looks clumpy. It would be appropriate to merge this section with the previous section 6 “Discussion and future perspectives”. 

Author Response

Dear Reviewer, thank you for your valuable comments, that will improve the quality of the manuscript. Here the change we made to match your review:

1) To improve readability, Section 5 “Other immunotherapeutic strategies for MSI-H cancers” would benefit from being divided into subsections.

Answer: we divided the section into subsections and we added table 2 to summarize it, as requested by another reviewer.

2) Section 7 “Conclusions” looks clumpy. It would be appropriate to merge this section with the previous section 6 “Discussion and future perspectives”. 

Answer: we eliminated the section "Conclusions" and merged its content in the previous section.

Tahnk you.

Sincerely

Round 2

Reviewer 1 Report

Comments and Suggestions for Authors The review has improved consideraly